# Long-Term Outcomes of Esophageal Squamous Neoplasia with Muscularis Mucosa Involvement after Endoscopic Submucosal Dissection

**DOI:** 10.3390/biomedicines12081660

**Published:** 2024-07-25

**Authors:** Chen-Huan Yu, Yueh-Hung Chou, Tze-Yu Shieh, Chao-Yu Liu, Jiann-Ming Wu, Chen-Hsi Hsieh, Tzong-Hsi Lee, Chen-Shuan Chung

**Affiliations:** 1Division of Gastroenterology and Hepatology, Department of Internal Medicine, Far Eastern Memorial Hospital, New Taipei City 220, Taiwan; michaelyu1004@gmail.com (C.-H.Y.); thleekimo@yahoo.com.tw (T.-H.L.); 2Department of Anatomical Pathology, Far Eastern Memorial Hospital, New Taipei City 220, Taiwan; komet1414@gmail.com; 3Division of Gastroenterology, Department of Internal Medicine, Mackay Memorial Hospital, Taipei City 104, Taiwan; range0425@gmail.com; 4Division of Thoracic Surgery, Department of Surgery, Far-Eastern Memorial Hospital, New Taipei City 220, Taiwan; chaoyuliu6@gmail.com; 5Division of General Surgery, Department of Surgery, Far Eastern Memorial Hospital, New Taipei City 220, Taiwan; klatskin@mail.femh.org.tw; 6Division of Radiation Oncology, Department of Radiology, Far Eastern Memorial Hospital, New Taipei City 220, Taiwan; chenciab@gmail.com; 7School of Nursing, Yuan Ze University, Taoyuan City 320, Taiwan; 8School of Medicine, National Yang Ming Chiao Tung University, Taipei City 112, Taiwan; 9Taiwan Association for the Study of Intestinal Diseases (TASID), Taoyuan City 333, Taiwan; 10College of Medicine, Fu Jen Catholic University, New Taipei City 242, Taiwan

**Keywords:** esophageal squamous cell carcinoma, endoscopic submucosal dissection, second primary tumor, head and neck cancer, expanded indication

## Abstract

Ambiguity exists over treatment and surveillance strategies after endoscopic submucosal dissection (ESD) for esophageal squamous cell neoplasia (ESCN) with unfavorable histologic features. This study investigated the long-term outcomes of ESD in high-risk ESCN patients. We retrospectively included early ESCN patients treated with ESD at two medical centers in Taiwan between August 2010 and December 2023. Demographic, endoscopic and pathological data were collected. Among 146 patients (mean age 59.17 years) with 183 lesions, 73 (50%) had a history of head and neck cancer (HNC). En bloc and R0 resections were achieved in 100% and 95.6% of the lesions, respectively. The 5-year overall survival (OS), disease-specific survival (DSS) and local recurrence rates were 42.7%, 94.7% and 11%. R0 resections were significantly associated with recurrence in a univariate analysis (HR: 0.19, 95% CI: 0.06–0.66, *p* = 0.008). Alcohol abstinence was independently associated with lower recurrence (HR: 0.34, 95% CI: 0.16–0.73, *p* = 0.006). Patients with pT1a-MM (muscularis mucosa invasion) had comparable OS (*p* = 0.82), DSS (*p* = 0.617) and recurrence (*p* = 0.63) rates to those with pT1a-EP/LPM (epithelium/lamina propria invasion). The long-term outcomes of ESCN patients after ESD for expanded indications were satisfactory. ESD could be considered in selected ESCN patients involving the muscularis mucosa, notably among high-risk HNC patients.

## 1. Introduction

Although the overall prognosis of esophageal squamous cell carcinoma (ESCC) is poor, with 5-year overall survival (OS) rates ranging from 15% to 25% [1], early esophageal squamous cell neoplasia (ESCN) is considered a curable disease [2,3]. The reported 5-year OS rate for patients diagnosed with early ESCC is approximately 80% following esophagectomy [4]. Endoscopic submucosal dissection (ESD) is an advanced endoscopic technique that was introduced to manage early gastrointestinal tract neoplasia in the 1990s. For selected patients with early ESCN, ESD has emerged as an alternative minimally invasive treatment option, with comparable outcomes to surgical esophagectomy [5]. A real-world cohort study with a 21-month follow-up of patients with early ESCN found no statistically significant differences in all-cause mortality (7.4% vs. 10.9%; *p* = 0.209) and the rate of cancer recurrence or metastasis (9.1% vs. 8.9%; *p* = 0.948) between ESD and esophagectomy [6]. Moreover, the authors reported a lower risk of non-fatal severe adverse events in the patients who underwent ESD (15.2% vs. 27.7%; *p* = 0.001) [6].

Favorable long-term outcomes have been reported in early ESCN patients undergoing ESD, particularly when the invasion is limited to the epithelium/lamina propria/muscularis mucosa (pT1a-EP/LPM/MM) or involves the superficial submucosa limited to 200 μm (pT1b-SM1), with 5-year OS rates of 90.5% to 95.1% for pT1a–EP/LPM, 71.1% to 84.2% for pT1a-MM, and 70.8% for pT1b-SM1, respectively [7,8,9,10,11,12]. However, these data were mostly obtained from Japanese cohorts, with only a few from Western countries. Furthermore, many of these studies excluded individuals with a history of synchronous second primary cancer [7,8,9].

In Taiwan, the rate of synchronous and metachronous second primary ESCN has been reported to range from 15.2 to 23.3% and 11.4% in head and neck cancer (HNC) patients, respectively [13,14,15]. Worse OS has been demonstrated in HNC patients with synchronous ESCN compared to those without [16]. According to recent studies, endoscopic screening for second primary esophageal cancer in patients with oral cavity and hypopharyngeal cancer can significantly enhance early detection and decrease all-cause mortality [17,18]. Nevertheless, long-term outcome data for ESD in patients with second primary esophageal neoplasia and HNC are still insufficient [12]. Therefore, this study aimed to examine the long-term outcomes of ESD for second primary ESCN among high-risk patients, particularly focusing on expanded indications.

## 2. Materials and Methods

### 2.1. Study Design and Patient Enrollment

This retrospective cohort study enrolled patients with early ESCN treated with ESD between August 2010 and December 2023 at two medical centers in Taiwan. All the patients met the inclusion criteria, which were (1) histologically confirmed low-grade intraepithelial neoplasia, high-grade intraepithelial neoplasia (HGIN) of the esophagus, or early ESCC including pT1a–EP/LPM, pT1a-MM and pT1b-sm1/sm2; (2) absence of lymph nodes or distant metastases in radiological investigations; and (3) no history of prior chemotherapy or radiation therapy for ESCC. Short-term and long-term outcome data were collected from medical records, and any missing information was obtained through telephone conversations with the patients or their family. Alcohol abstinence is defined as having abstained from alcohol for at least one year after ESD and continuing to abstain until the last follow-up date. This study was approved by the Institutional Review Board of Far Eastern Memorial Hospital (FEMH-106090-E), and the requirement for written informed consent was waived because of the absence of identifiable information in the data analyzed.

### 2.2. ESD Procedure and Surveillance Methods

All enrolled the patients underwent magnifying endoscopy with narrow-band imaging (ME-NBI) and optional chromoendoscopy using 2% Lugol’s solution to evaluate the intraepithelial papillary capillary loops and Lugol-voiding areas [19,20]. Overnight fasting and prophylactic antibiotics were given, and the patients were placed in a supine or left decubitus position under intravenous sedation or general anesthesia. ESD with mucosal marking and incision, followed by submucosal dissection and specimen retrieval, was performed using dedicated endoknives and accessories. The resected specimens were sliced into 2 mm sections and fixed in formalin using standard methods following the Japan Esophageal Society recommendations, and all specimens were reviewed by experienced pathologists [21]. The pathological invasion depth, lymphovascular and perineural invasion, histological subtype and differentiation degree, and vertical and lateral margins were evaluated. All the ESD procedures were performed by Dr. Chen-Shuan Chung at Far Eastern Memorial Hospital and Dr. Tze-Yu Shieh at Mackay Memorial Hospital. Both endoscopists have performed over 500 cases of ESD.

All the patients underwent computed tomography (CT) scans before ESD to exclude lymph node dissemination. Pre-ESD depth evaluation was all determined by intraepithelial papillary capillary loops under ME-NBI [19]. Endoscopic ultrasound was performed at the discretion of endoscopists if they suspected submucosa invasion. The first follow-up endoscopy in all cases was performed at 3 to 6 months after ESD. Subsequent endoscopic surveillance was repeated every 6 months. CT was performed for follow-up in all the patients with components of invasive carcinoma in resected specimens.

### 2.3. Definition of Outcomes

The primary objectives of this study were to assess long-term outcomes, including survival and recurrence, among high-risk early ESCN patients after ESD. OS, disease-free survival (DFS) and disease-specific survival (DSS) were compared among patients with different invasion depths and those with or without HNC. OS was defined as the time from ESD to disease from any cause. DFS was defined as the time from ESD to esophageal cancer recurrence or death from any cause. DSS was defined as the duration from ESD to death specifically attributed to the ESCN.

Local recurrence and metastatic recurrence were evaluated by endoscopy, CT and/or positron emission tomography. Local recurrence was defined as the development of ESCN at the ESD site of the primary lesion, while metastatic recurrence was defined as a relapse involving other organs or lymph nodes. Metachronous recurrence was defined as the development of an ESCN at a location different from the primary lesion site over 6 months after the index ESD. Furthermore, we also investigated potential correlations between the occurrence of any recurrence and baseline characteristics of the patients as well as lesions.

Short-term outcomes, including R0 resection and complete local remission (CLR), were reported as secondary objectives of this study. CLR was defined as R0 resection plus normal first follow-up endoscopy. In terms of adverse events from ESD procedures, esophageal stricture was defined as the narrowing of the lumen that a standard adult gastroscope could not pass or required endoscopic dilation. Post-ESD bleeding was defined as bleeding occurring after the ESD, characterized by a decrease in hemoglobin levels > 2 g/dL, the presence of hematemesis, or melena or confirmation of bleeding through endoscopic, radiological or surgical evidence. Esophageal perforation was defined as the presence of a visible defect in the esophageal wall exposing the mediastinal cavity.

### 2.4. Statistical Analysis

Basic characteristics of the patients were evaluated via descriptive statistics. Discrete data are presented as numbers and percentages, and continuous variables are expressed as mean values (±standard deviation; SD). The chi-squared test was used to evaluate categorical variables. Two-tailed *p*-values < 0.05 were used to determine statistical significance. A logistic regression analysis was conducted to evaluate the relationship between risk factors and outcomes. Univariate and multivariate Cox regression analyses were used to identify risk factors associated with time-to-event outcomes. Variables from a univariate analysis with *p*-values < 0.10 were included in the multivariate model. A survival analysis was performed using the Kaplan–Meier method along with a log-rank test to assess the statistical significance of the differences in survival curves among different groups. All statistical analyses were performed with SPSS for Windows, version 26.0 (IBM Corp., Armonk, NY, USA). The study results were reviewed by expert biostatistician Dr. Hsiu-Jung Liao.

## 3. Results

### 3.1. Patient, Endoscopic and Pathologic Characteristics

A total of 183 lesions in 146 patients were studied. Table 1 summarizes the demographic characteristics of the study population. The mean (±SD) patient age was 59.17 (±9.45) years, and 89% (130/146) of the patients were male. This study included 132 (90.4%) patients who were alcohol drinkers, 68.9% of whom abstained from alcohol prior to enrollment. A total of 73 (50%) patients had a history of HNC. Table A1 details the location and stage of HNC in this study. The endoscopic and pathological characteristics are also summarized in Table 1. Among the lesions, 43 (23.5%) extended to more than three-fourths of the circumference. The pathologic stage was low-grade intraepithelial neoplasia in 16 (8.7%) lesions, HGIN/T1am1 in 90 (49.1%), T1am2 in 10 (5.5%), T1am3 in 42 (23%), T1bsm1 in 10 (5.5%), and T1bsm2 in 15 (8.2%) lesions. The frequencies of lymphovascular invasion (LVI) and perineural invasion were 14/183 (7.6%) and 3/183 (1.6%), respectively.

### 3.2. Short-Term Outcomes and ESD-Related Complications

The short-term outcomes and complications are shown in Table 2. En bloc resection, R0 resection and CLR were achieved in 183 (100%), 175 (95.6%) and 161 (88%) of the 183 lesions, respectively. Post-ESD strictures developed in 18 (9.8%) of the 146 patients, with no cases developing bleeding or perforation and no cases of procedure-related mortality.

### 3.3. Long-Term Outcomes

The long-term outcomes are presented in Table A2. The mean (±SD) follow-up time was 37.05 months (±26.79), and the 5-year OS and DSS rates were 42.7% and 94.7%, respectively. Over the 10-year follow-up period, the cumulative recurrence rate was 20.5%. Specifically, the 5-year local recurrence rate was 11%, and metachronous recurrence occurred in 14 cases (9.5%). The 5-year cumulative incidence of metastatic recurrence was 2.8% in the pT1a-EP/LPM group and 7.8% in the pT1a-MM group.

Figure 1a,b show that the pathologic stage had statistically significant impacts on OS (*p* = 0.022) and DSS (*p* = 0.007). In the HGIN/pT1a–EP/LPM group, the 3-year and 5-year OS rates were 67.5% and 50%, respectively, compared to 68.8% and 52% in the pT1a-MM group (*p* = 0.82, Figure 2a). The Kaplan–Meier curve showed no statistically significant difference in OS when comparing the HGIN/pT1a–EP/LPM group with the pT1a-MM group without LVI (*p* = 0.793, Figure 2b). The 5-year DSS rates were 96.7% and 100% in the HGIN/pT1a–EP/LPM and pT1a-MM groups, respectively (*p* = 0.617, Figure 2c). In the patients with pT1a-MM, the presence of LVI did not affect OS (*p* = 0.96, Figure 2d). The causes of death in our study population are detailed in Table A3.

Table A4 provides the analysis of the impact of HNC on OS, DSS and DFS. The patients with HNC had a lower 5-year OS rate than those without HNC (adjusted odds ratio [aOR]: 0.31, 95% confidence interval [CI]: 0.12–0.82, *p* = 0.018). In addition, a higher DFS was also found in the patients without HNC in Kaplan–Meier curve (*p* = 0.013, Figure A1). No significant difference was observed in the recurrence rate among different locations of HNC (*p* = 0.915, Figure A2, Figure A3 and Figure A4).

The descriptive data of the patients who underwent additional therapy are shown in Table A5. Following ESD, a total of 19 patients received additional therapy, with 13 patients (68.4%) undergoing concurrent chemoradiotherapy. The primary reason for additional therapy was failure to achieve R0 resection (8 patients, 42.1%), followed by deep submucosal invasion (5 patients, 26.3%).

### 3.4. Predictors Associated with Recurrence

The results of univariate and multivariate analyses for the factors associated with recurrence after ESD are demonstrated in Table 3. The patients who achieved R0 resection and CLR and those who abstained from alcohol consumption had a lower rate of recurrence. The multivariate analysis showed that alcohol abstinence was the only independent factor associated with a decreased recurrence rate (HR: 0.34, 95% CI: 0.16–0.73, *p* = 0.006). Kaplan–Meier curve demonstrated no statistically significant difference in recurrence between the patients with HGIN/pT1a–EP/LPM and those with pT1a-MM (*p* = 0.63) (Figure 3a,b).

## 4. Discussion

In this study, we investigated the short-term outcomes, complications and long-term outcomes of patients with early ESCN after ESD with expanded indications, and we observed favorable 5-year DSS (94.7%), en bloc resection (100%) and R0 resection (95.6%) rates following ESD for early ESCN in high-risk HNC patients. Regarding survival and disease recurrence, we found no significant difference between the patients with pT1a–MM and those with pT1a–EP/LPM. Our results suggest that ESD without additional therapy may be a reasonable treatment option for patients with ESCN involving pT1a-MM.

In contrast to the significant perioperative mortality rate associated with esophagectomy (previously reported as ranging from 1.5% to 12.5%) [6,22,23], ESD, as a minimally invasive procedure, has been associated with fewer perioperative comorbidities in selected patients with early ESCN [7,8,9,10]. Moreover, a previous study found that patients who underwent ESD had a lower risk of non-fatal severe adverse events compared to those who underwent esophagectomy for early ESCN (15.2% vs. 27.7%; *p* = 0.001) [6]. In the present study, ESD was not associated with any cases of perioperative mortality, and the post-ESD stricture rate was only 9.8%, which is consistent with the rates of 6% to 18.1% reported in large cohort studies with adequate stricture prevention [7,8,10,12,24]. Other advantages of ESD include the high en bloc and R0 resection rates, which have been reported to reach 95% to 100% and 84.5% to 99.3%, respectively, in Asian studies [24,25,26,27]. In comparison, reports from Western countries indicate varying R0 resection rates for ESD in early ESCN, ranging from 69.6% to 96.7% [11,28,29]. A previous prospective study reported a higher en bloc resection rate with ESD compared with endoscopic mucosal resection for early ESCN (100% vs. 53.3%; *p* = 0.009) [30]. In our study, we achieved favorable en bloc and R0 resection rates of 100% and 95.6%, respectively, consistent with previous research. In addition, the CLR rate in our study was 88%. Only a few studies have reported the CLR rate, including one study with a rate of 95.7% for ESD in cases with early ESCN [31] and another with a rate of 96.6% for Barrett’s esophagus [32].

Regarding disease recurrence, a real-world cohort study comparing ESD with esophagectomy in selected patients with pT1a-LPM/MM and pT1b-SM (invading the submucosa layer) found no statistically significant difference in cancer recurrence or metastasis during the 21-month follow-up period (9.1% vs. 8.9%; *p* = 0.948) [6]. This suggests favorable long-term outcomes of ESD in selected early ESCN patients. In the present study, the only independent factor associated with disease recurrence was alcohol abstinence, aligning with findings from a previous prospective cohort study [33]. In the multivariate analysis, neither R0 resection nor CLR was associated with a higher recurrence rate. This may be because all the patients who did not achieve R0 resection after ESD received additional therapy. Our 5-year local recurrence rate of 11% is in line with the range of 3.9% to 16.8% reported in patients without HNC in large Japanese cohort studies [7,8,9]. Furthermore, the high metachronous recurrence rate of 17.9% after ESD in HNC patients reported in previous research was not observed in our study [12]. This finding may be associated with our high R0 resection rate and the significant prevalence of alcohol abstinence after education, notably among the patients with HNC.

Previous studies have reported cumulative 5-year metastatic recurrence rates for patients diagnosed with pT1a-EP/LPM and pT1a-MM ranging between 0 and 0.5% and between 3.3 and 8.7% [7,9,10], respectively, which are consistent with our results. Due to an elevated rate of lymph node metastasis, the Japan Esophageal Society guidelines in 2017 recommended additional treatment after ESD for pT1a-MM ESCC patients [3]. However, an increasing number of studies suggests a positive prognosis in pT1a-MM patients after ESD even without additional therapy [10,34]. A recent study conducted in Japan found that ESD for pT1a-MM patients without LVI exhibited comparable DSS to pT1a-EP/LPM patients (99.3% vs. 96.6%, *p* = 0.118), with no significant differences in the cumulative incidence of metastatic and local recurrence (*p* = 0.121 and *p* = 0.455, respectively) [10]. Our data support these results, in that we observed no significant differences in OS, DSS and post-ESD recurrences between the patients with pT1a-MM without additional therapy and those with pT1a-EP/LPM without additional therapy (Figure 2a,d and Figure 3b). Even when considering the presence or absence of LVI, no statistically significant differences were observed between the two groups (Figure 2b,c). Our findings suggest that expanding the indications for ESD without additional therapy to include pT1a-MM patients without LVI seems reasonable and consistent with previous research [10]. Further research is required to confirm the clinical outcomes of pT1a-MM patients after ESD, and appropriate follow-up strategies need to be established.

The favorable outcomes of ESD for second primary early ESCN in the high-risk HNC patients in this study are also noteworthy. Even though the 5-year OS rate in our patients was only 48%, which is lower than the 95.1% to 99% reported in Japanese cohorts [8,9], the 5-year DSS reached 94.7%, which is not far from the 99.1% to 100% reported previously [8,9]. The lower OS in our cohort may be due to the significant proportion of patients with HNC (50%), among whom 53.4% were at stage IV (Table A1), with 50% of deaths related to HNC (Table A3). Similar findings were reported in a retrospective study involving 167 patients, 61.7% of whom were diagnosed with HNC [12]. The 10-year disease-related survival rate was more than 90%, whereas the OS rate was less than 70% [12]. We previously reported that screening and surveillance of ESCN by ME-NBI can provide opportunities for early treatment and even improve the survival of hypopharyngeal cancer patients [18]. Given the positive short-term and long-term outcomes observed in this study, we believe that ESD with expanded indications for second primary ESCN can provide clinical benefits with a favorable DSS rate. In addition, in our previous case–control study, tumors located in the hypopharynx and pharynx were associated with a higher risk of developing second primary ESCN in HNC patients (aOR 4.52, 95% CI: 1.46–13.99, *p* = 0.009 and aOR 5.70, 95% CI: 1.08–29.99, *p* = 0.04) [13]. However, the impact of HNC tumor sites on recurrence after ESD for second primary early ESCN remains unclear. In the present study, although the presence of HNC was not significantly associated with recurrence (Table 3), there appeared to be a trend suggesting a higher incidence of recurrence in laryngeal and hypopharyngeal cancers compared to cancers of the oral cavity and oropharynx (Figure A2, Figure A3 and Figure A4). More studies are needed to determine the relationship between HNC tumor site and recurrence of second primary early ESCN after ESD.

Some limitations of this study should be noted. First, this study adopted a retrospective design, which may have introduced potential selection and measurement bias. Second, the limited number of pT1a-MM patients with concurrent LVI in this study undermines the reliability of assessing the statistical significance of LVI on clinical outcomes in the pT1a-MM subgroup. Third, the heterogeneity of the study population, including patients with HNC from various primary sites and those without HNC, may impact the generalizability of the findings.

## 5. Conclusions

In summary, our results showed favorable short-term and long-term outcomes of ESD for second primary early ESCN in patients with HNC. We also found that ESD in patients with pT1a-MM yielded comparable results to those with pT1a-EP/LPM in terms of survival and disease recurrence. This highlights the potential for expanding the indications for ESD in managing pT1a-MM patients without additional therapy. Further research is necessary to validate these findings.

## Figures and Tables

**Figure 1 biomedicines-12-01660-f001:**
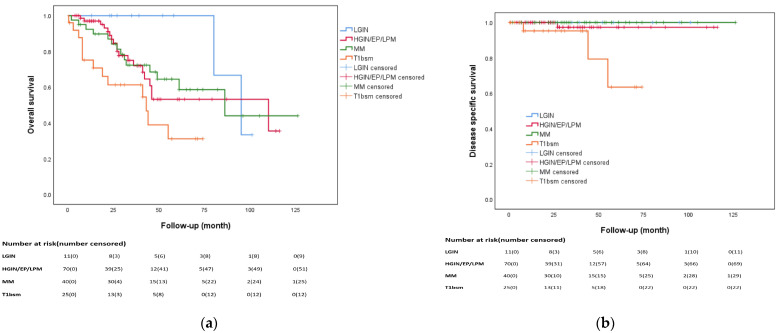
Kaplan–Meier curve for the impact of pathologic stage on overall survival and disease-specific survival: (**a**) Statistically significant difference was observed (*p* = 0.022) for overall survival; (**b**) Statistically significant difference was observed (*p* = 0.007) for disease-specific survival.

**Figure 2 biomedicines-12-01660-f002:**
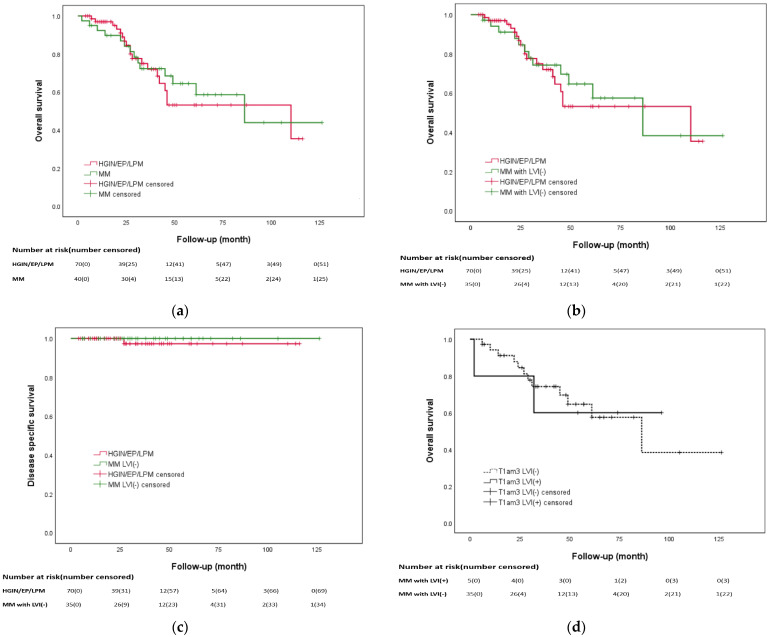
Kaplan–Meier curve of patients with HGIN/pT1a–epithelium/lamina propria mucosa (HGIN/EP/LPM) without additional therapy and patients with pT1a–muscularis mucosa (MM), lymphovascular invasion (LVI) negative and no additional therapy: (**a**) overall survival between patients with HGIN/EP/LPM and those with MM (*p* = 0.82); (**b**) overall survival between HGIN/EP/LPM patients and MM patients without LVI (*p* = 0.793); (**c**) disease-specific survival between HGIN/EP/LPM patient and MM patients without LVI (*p* = 0.617); (**d**) overall survival with respect to the presence of LVI in T1am3 (pT1a–muscularis mucosa) patients (*p* = 0.96).

**Figure 3 biomedicines-12-01660-f003:**
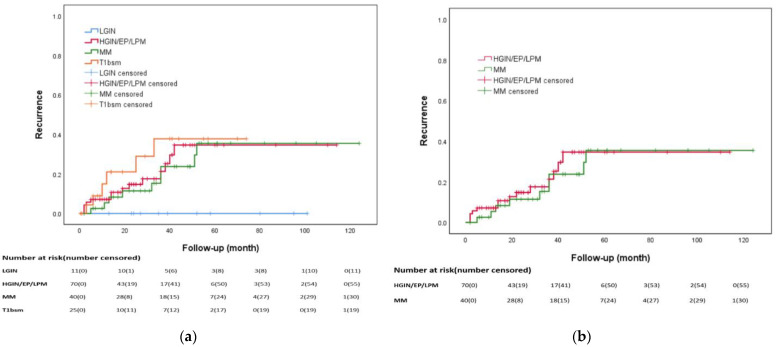
Kaplan–Meier curve for cumulative incidence of recurrence: (**a**) No statistically significant difference was observed (*p* = 0.229) from different pathologic stages; (**b**) No statistically significant difference was observed (*p* = 0.63) in HGIN/EP/LPM patients without additional therapy and MM patients without additional therapy.

**Table 1 biomedicines-12-01660-t001:** Clinical characteristics of 183 lesions in 146 patients.

Variable	Lesion No. (%)
Patient characteristics Age (mean ± SD), years	
	59.17 ± 9.45
Sex	
Male	130 (89)
Female	16 (11)
Smoking	125 (85.6)
Alcohol drinker	132 (90.4)
Betel nut chewing	64 (43.8)
Smoking cessation	48 (38.4)
Alcohol abstinence	91 (68.9)
Betel nut cessation	45 (70.3)
HN cancer history	73 (50)
Endoscopic characteristics	
Tumor location	
Upper third of the esophagus	24 (13.1)
Middle third of the esophagus	102 (55.7)
Lower third of the esophagus	57 (31.1)
Endoscopic tumor size (mean ± SD), cm	2.48 ± 1.90
Circumference of the tumor	
<1/2	71 (38.8)
<3/4	140 (76.5)
≥3/4	43 (23.5)
JES type ^1^	
B1	140
LGIN/HGIN/T1am1/T1am2	109 (77.9)
T1am3/T1bsm1	28 (20)
T1bsm2	3 (2.1)
B2	43
LGIN/HGIN/T1am1/T1am2	7 (16.3)
T1am3/T1bsm1	24 (55.8)
T1bsm2	12 (27.9)
Pathological characteristics	
Histological subtype	
Well differentiated (G1)	18 (23.3)
Moderately differentiated (G2)	55 (71.4)
Poorly differentiated (G3)	4 (5.1)
LVI	14 (7.6)
Perineural invasion	3 (1.6)
Pathologic stage	
LGIN	16 (8.7)
HGIN/T1am1	90 (49.1)
T1am2	10 (5.5)
T1am3	42 (23)
T1bsm1	10 (5.5)
T1bsm2	15 (8.2)

SD: standard deviation; HN cancer: head and neck cancer; LVI: lymphovascular invasion. ^1^ The JES type classification is based on the magnifying endoscopic classification of the Japan Esophageal Society [19].

**Table 2 biomedicines-12-01660-t002:** Short-term outcomes and complications of patients treated with ESD.

Variable	Results
En bloc resection, no. of lesions (%) R0 resection, no. of lesions (%)	183 (100)
	175 (95.6)
Complete local remission, no. of lesions (%)	161 (88)
Overall post-ESD AE	18 (9.8)
Stricture	18 (9.8)
Bleeding	0
Perforation	0

ESD: endoscopic submucosal dissection; AE: adverse event; SD: standard deviation.

**Table 3 biomedicines-12-01660-t003:** Univariate and multivariate analysis of factors associated with recurrence.

	Recurrence	Non-Recurrence	Univariate		Multivariate	
	(n = 30)	(n = 116)	*p*-Value	HR (95% CI)	*p*-Value	HR (95% CI)
Males	27 (90%)	103 (89%)	0.67	1.30 (0.39–4.30)		
Age (mean ± SD), years	56.63 ± 9.59	59.83 ± 9.34	0.38	0.98 (0.94–1.02)		
Smoking	26 (87%)	99 (85%)	0.94	1.04 (0.36–3.00)		
Alcohol	30 (100%)	102 (88%)				
Betel nut	15 (50%)	49 (42%)	0.35	1.41 (0.68–2.93)		
Smoking cessation	13 (50%)	35 (35%)	0.65	1.20 (0.54–2.66)		
Alcohol abstinence	12 (40%)	79 (78%)	0.002	0.30 (0.14–0.63)	0.006	0.34 (0.16–0.73)
Betel nut cessation	11 (73%)	34 (69%)	0.95	1.04 (0.33–3.33)		
HN cancer	16 (53%)	57 (49%)	0.81	1.09 (0.53–2.26)		
Oral cavity cancer	3 (19%)	15 (26%)	0.54	0.67 (0.19–2.39)		
HPC and laryngeal cancer	10 (63%)	30 (53%)	0.67	1.25 (0.44–3.51)		
LVI	5 (17%)	9 (8%)	0.16	1.99 (0.76–5.24)		
Perineural invasion	0	3 (3%)				
T1am3/T1bsm	16 (53%)	49 (42%)	0.55	1.25 (0.60–2.60)		
T1bsm	6 (20%)	19 (16%)	0.27	1.67 (0.68–4.10)		
Histology subtype G2/G3	13 (62%)	45 (76%)	0.24	0.57 (0.23–1.45)		
Histology subtype G3	0	4 (3%)				
Endoscopy tumor size (mean ± SD), cm	3.60 ± 2.01	3.23 ± 2.18	0.44	1.06 (0.92–1.21)		
Circumference over 3/4	8 (27%)	30 (26%)	0.55	1.29 (0.57–2.91)		
Circumference over 1/2	23 (77%)	78 (67%)	0.16	1.84 (0.78–4.32)		
R0 resection	26 (87%)	112 (97%)	0.008	0.19 (0.06–0.66)	0.70	0.74 (0.16–3.42)
CLR	23 (77%)	103 (89%)	0.003	0.27 (0.11–0.63)	0.12	0.42 (0.14–1.25)

HR: hazard ratio; CI: confidence interval; SD: Standard deviation; HN cancer: head and neck cancer; HPC: hypopharyngeal cancer; LVI: lymphovascular invasion; CLR: complete local remission.

## Data Availability

The datasets analyzed during this study are available from the corresponding author on reasonable request due to privacy and ethical reasons.

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
