# Peer review of "Long-Term Outcomes of Esophageal Squamous Neoplasia with Muscularis Mucosa Involvement after Endoscopic Submucosal Dissection"

_biomedicines, 2024, doi:10.3390/biomedicines12081660_

Round 1

Reviewer 1 Report

Comments and Suggestions for Authors

Congratulations to Yu et al. on this interesting manuscript. The authors investigated the usefulness of Endoscopic Submucosal Dissection (ESD) for early esophageal squamous neoplasia with muscularis mucosa involvement. Overall, the study is well-written. However, there are a few points that need to be addressed to improve the manuscript:

Title:

The title is too long. Consider shortening it for clarity and impact.

Introduction:

The authors state that their objective was to evaluate the long-term outcomes of ESD for second primary esophageal squamous cell neoplasia (ESCN) among high-risk patients. However, the study also investigates the short-term outcomes of ESD. I suggest redefining the objective to clearly establish primary and secondary goals.

Methods:

Clarify how patients were evaluated before ESD. Did every patient undergo a CT scan, PET scan, and echoendoscopy? It is important to know how the authors ruled out lymph node dissemination.

Specify who performed the ESD procedures.

Results:

Table 2 currently depicts both short- and long-term outcomes, which is confusing. I recommend using this table exclusively for short-term outcomes and presenting the long-term outcomes using Kaplan-Meier figures.

Ensure that the Kaplan-Meier curves include the number at risk.

In Table 3, it is unclear how recurrence was assessed. It appears that the authors did not use recurrence as a time-to-event measure. For time-to-event outcomes like recurrence, the associated measure is usually the hazard ratio, not the odds ratio. It is difficult to assess recurrence without considering the follow-up period, as patients might have varying follow-up durations.

Conclusion:

The conclusion should focus on addressing the study’s objectives, including both primary and secondary outcomes, rather than summarizing the results.

Author Response

Comments 1:

The title is too long. Consider shortening it for clarity and impact.

Response 1:

Thank you for the comments. Based on your suggestions, we have revised the title to: "Long-Term Outcomes of Esophageal Squamous Neoplasia with Muscularis Mucosa Involvement after Endoscopic Submucosal Dissection."

Comments 2:

The authors state that their objective was to evaluate the long-term outcomes of ESD for second primary esophageal squamous cell neoplasia (ESCN) among high-risk patients. However, the study also investigates the short-term outcomes of ESD. I suggest redefining the objective to clearly establish primary and secondary goals.

Response 2: Thank you for the comments. Although our primary objective was to evaluate the long-term outcomes of ESD for second primary esophageal squamous cell neoplasia (ESCN) among high-risk patients, we also considered that short-term outcomes, such as R0 resection rate, complete local remission (CLR), and complications, could impact the long-term results. Therefore, we reported these short-term outcomes as secondary objectives of the study. The primary and secondary objectives have been clarified in the “Definition of Outcomes”section.

Comments 3:

Clarify how patients were evaluated before ESD. Did every patient undergo a CT scan, PET scan, and echoendoscopy? It is important to know how the authors ruled out lymph node dissemination.

Response 3:

Based on your comment, we have added the following information to the "ESD Procedure and Surveillance Methods" section: "All patients underwent computed tomography scans before ESD to exclude lymph node dissemina-tion. Pre-ESD depth evaluation was all determined by intraepithelial papillary capillary loops under magnifying endoscopy with narrow-band imaging. Endoscopic ultrasound was done at the discretion of endoscopists if there is suspicious of submucosa invasion."

Comments 4:

Specify who performed the ESD procedures.

Response 4:

Thank you for the comments. Based on your suggestion, we have added the following information to the "ESD Procedure and Surveillance Methods" section: " All ESD procedures were performed by Dr. Chen-Shuan Chung at Far Eastern Memo-rial Hospital and Dr. Tze-Yu Shieh at Mackay Memorial Hospital. Both endoscopists have performed over 500 cases of ESD."

Comments 5:

Table 2 currently depicts both short- and long-term outcomes, which is confusing. I recommend using this table exclusively for short-term outcomes and presenting the long-term outcomes using Kaplan-Meier figures.

Response 5:

Thank you for the comments. Following your suggestion, we retained the short-term outcomes in Table 2 and use Kaplan-Meier curves to present the long-term outcomes. However, since long-term outcomes such as overall survival (OS), disease-specific survival (DSS) are our primary outcomes, detailed information on these will still be shown in the Appendix (Table A2) for the readers' reference.

Comments 6:

Ensure that the Kaplan-Meier curves include the number at risk.

Response 6:

Thank you for your comments. Information on the number at risk has been added to the Kaplan-Meier curves (Figures 1-3).

Comments 7:

In Table 3, it is unclear how recurrence was assessed. It appears that the authors did not use recurrence as a time-to-event measure. For time-to-event outcomes like recurrence, the associated measure is usually the hazard ratio, not the odds ratio. It is difficult to assess recurrence without considering the follow-up period, as patients might have varying follow-up durations.

Response 7: Thank you for the comments. Cox regression analyses were used to identify risk factors associated with time-to-event outcomes, with the hazard ratios and 95% CIs shown in Table 3. Revised information is presented in the “Statistical Analysis” section as follows: “Logistic regression analysis was conducted to evaluate the relationship between risk factors and outcomes. Univariate and multivariate Cox regression analyses were used to identify risk factors associated with time-to-event outcomes.”

Comments 8:

The conclusion should focus on addressing the study’s objectives, including both primary and secondary outcomes, rather than summarizing the results. 

Response 8:  Thank you for the comments. Our study found favorable results for both primary and secondary outcomes. Notably, pT1a-MM invasion also showed positive long-term outcomes compared to other invasion depths. Thus, we summarized these findings and suggested expanding the indications for ESD in the conclusion.

Reviewer 2 Report

Comments and Suggestions for Authors

This article is well written but the subject in general is not new since lot of similar studies was done in eastern and western countries with similar results as of the favorable outcome of ESD in early ESCN ; the originality is the results in termes of survival and disease recurrence observed for patients with MM.

here are some minor comments:

introduction: what is the rate of metachronous second primary in Taiwan?

Materials and méthodes:

2.3: define  synchronous recurrence 

Results

table 1: clarify in legend what are JES types B1 and B2

table 2: column 1 : erase Lesions No (%) since it is stated in the title of column 2

Author Response

Comments 1:

What is the rate of metachronous second primary in Taiwan?

Response 1:

Thank you for the comments. Few studies have discussed metachronous ESCN in head and neck cancer patients. In a prospective study conducted in southern Taiwan, 11.4% (20/175) of patients with head and neck squamous cell carcinoma were found to have metachronous ESCN (Cancers (Basel). 2020 Dec 18;12(12):3832.).

The manuscript was revised in the 'Introduction' section as follows: “In Taiwan, the rate of synchronous and metachronous second primary ESCN has been reported to range from 15.2~23.3% and 11.4% in head and neck cancer (HNC) patients , respectively.”

Comments 2:

define synchronous recurrence

Response 2:

Thank you for the comments. Actually, we did not use the term "synchronous recurrence" in our study. We defined "local recurrence" as the development of ESCN at the ESD site of the primary lesion. "Metachronous recurrence" was defined as the development of ESCN at a location different from the primary lesion site over 6 months after the index ESD. There was no patient in our cohort who developed ESCN at a location different from the primary lesion site within 6 months after ESD.

Comments 3:

table 1: clarify in legend what are JES types B1 and B2

Response 3:

The JES type classification is based on the magnifying endoscopic classification of the Japan Esophageal Society, which is derived from intrapapillary capillary loop findings (Esophagus. 2017;14(2):105-112). The reference has been added to the legend.

Comments 4:

table 2: column 1 : erase Lesions No (%) since it is stated in the title of column 2

Response 4:

Thank you for the comments. “Lesions No (%)” has been replaced with “Results”.

Round 2

Reviewer 1 Report

Comments and Suggestions for Authors

I would like to congratulate the authors on their thorough and well-executed study. The authors have addressed all the comments and suggestions, resulting in a clear and impactful manuscript. They have shortened the title for clarity and impact, redefined the objectives, and clarified the methodology. The authors have also restructured Tables, separating short- and long-term outcomes in different Tables. Survival and recurrence were evaluated as time-to-event outcomes. These revisions have significantly improved the manuscript, and I recommend it be accepted for publication in its current form.

Reviewer 2 Report

Comments and Suggestions for Authors

All comments have been responded